# Knowledge, attitudes and practices towards emergency contraceptive pill use among women of reproductive age in Lira District, Northern Uganda: A cross-sectional study

Josephine Vanessa Nakalema[1], Marvin Musinguzi[2]*, Deo Benyumiza[3], Eustes Kigongo[4], Edward Kumakech[3], Rashida Namuwaya[3], Joshua Oryem Opido[5], Nicholas Damulira[1], Raymond Tumwesigye[6], Amir Kabunga[7], Marc Sam Opollo[1]

1 Department of Community Health, School of Public Health Lira University, Lira, Uganda, 2 Department of Health sciences, Faculty of Social Sciences, Tampere University, Tampere, Finland, 3 Department of Midwifery, Faculty of Nursing and Midwifery, Lira University, Lira, Uganda, 4 Departments of Environmental Health and Disease Control, Faculty of Public Health, Lira University, Lira, Uganda, 5 Department of Nursing, Faculty of Health Sciences, Mbarara University of science and Technology, Mbarara, Uganda, 6 Department of Psychiatry, Faculty of Medicine, Lira University, Lira, Uganda, 7 Uganda Technical College, Department of Water and Sanitation P.O.BOX 04, Lira, Uganda,

* mmusinguzimarvin@lirauni.ac.ug, marvin.musinguzi@tuni.fi

## Abstract

### Background

Emergency contraceptives (ECs) are used to prevent unintended pregnancy after unprotected sexual intercourse. In Uganda, unsafe abortion remains a major contributor to maternal mortality, particularly in resource-limited settings such as Northern Uganda. Greater use of emergency contraceptive pills could help reduce unintended pregnancies and their related consequences. However, information on women's knowledge, attitudes, and practices regarding emergency contraceptives remains limited. This study therefore assessed the knowledge, attitudes, and practices of women of reproductive age towards emergency contraceptives in Lira District.

### Method

This was a community-based cross-sectional study conducted between June and July 2023. A total sample size of 600 was estimated; however, 585 respondents were ultimately selected using multi-stage cluster sampling. Data were analyzed using SPSS version 26 at the univariate, bivariate, and multivariate levels.

### Results

The study had a response rate of 97.5% (585). Emergency contraceptive pill usage was at 30.6%. The mean age of participants was 28.01 ± 7.6 years. Majority of the respondents 374 (63.93%) had good knowledge about emergency contraceptive pills.

**Data availability statement:** All relevant data are within the manuscript and its Supporting Information files.

**Funding:** The author(s) received no specific funding for this work.

**Competing interests:** The authors have declared that no competing interests exist.

Most of the respondents 386 (65.98%) had negative attitudes towards emergency contraceptive pills. The factors associated with use of emergency contraceptive pills include: having poor knowledge about emergency contraception (AOR = 0.11, 95% CI 0.07–0.21, p < 0.001), having positive attitudes towards emergency contraceptive pills (AOR = 3.89, 95% CI 2.59–5.84, p < 0.001), and being unmarried (AOR = 2.85, 95% CI 1.83–4.43, p < 0.001).

## Conclusion

Emergency contraceptive pill utilization among women of reproductive age in Lira District was modest, with about three in ten women reporting use in the previous month. Utilization was significantly influenced by knowledge, attitudes, and marital status. These findings highlight the need for interventions that strengthen knowledge, address negative attitudes, and support appropriate use of emergency contraception.

## Background

Contraceptive use remains a significant global public health concern. In 2020, an estimated 1.9 billion women were of reproductive age (15–49 years) worldwide, of whom 1.1 billion had a need for family planning services [1]. Among these, 851 million were using modern contraceptive methods, while 85 million relied on traditional methods. Despite this progress, unmet need for contraception continues to result in approximately 74 million unintended pregnancies annually in low- and middle-income countries [2]. These unintended pregnancies contribute to about 25 million unsafe abortions and nearly 47,000 maternal deaths each year. Strengthening access to effective contraceptive methods, including emergency contraception, is therefore critical in preventing unintended pregnancies and reducing unsafe abortions [3].

Emergency contraceptives (ECs) are methods used to prevent pregnancy after unprotected sexual intercourse or contraceptive failure. They are most effective when used as soon as possible after intercourse but can be used up to five days later, depending on the method. The primary forms of emergency contraception include copper-bearing intrauterine devices (IUDs) and emergency contraceptive pills (ECPs) [4]. According to the World Health Organization (WHO), recommended ECP regimens include ulipristal acetate, levonorgestrel-only pills, and combined oral contraceptives (COCs) containing ethinyl estradiol and levonorgestrel [5]. These methods primarily work by delaying or inhibiting ovulation. In Uganda, commonly available emergency contraceptive pills include brands such as Postinor and Plan B, which are accessible through both public and private health facilities [6].

In Uganda, overall contraceptive prevalence remains low at 26%, compared to neighbouring countries such as Kenya, where prevalence stands at 53%. [7]. Emergency contraception was officially introduced by the Ministry of Health in November 1998 to improve reproductive health outcomes [8]. However, its utilization remains extremely low—0.1% among all women, 0.1% among married women, and 0.3% among sexually active unmarried women [7]. In contrast, studies among university

students have reported significantly higher usage rates, reaching up to 45% [5], suggesting disparities in awareness and access across population groups.

Despite policy provisions allowing trained Village Health Teams (VHTs) to distribute selected contraceptive methods, including emergency contraceptive pills, awareness of emergency contraception among women in Uganda remains low [6]. Only 37.7% of women report knowledge of ECPs, compared to much higher awareness of injectables (95.9%), IUDs (80.1%), and condoms (80.6%) [7]. The continued low uptake of emergency contraception contributes to Uganda's high rate of unsafe abortions, estimated at 54 per 1,000 women of reproductive age annually [9].

In Northern Uganda, where Lira District is located, the contraceptive nonuse rate is about 55%, whereas Central Uganda reports a lower contraceptive nonuse rate of 35% [10]. Contraceptive use is influenced by factors such as wealth index, number of living children, and level of education [10]. Despite relatively higher general contraceptive use in Northern Uganda, there is limited documented evidence specifically addressing emergency contraceptive use in Lira District.

Therefore, this study aims to assess the knowledge, attitudes, and practices of women of reproductive age regarding emergency contraceptives in Lira District. The findings may provide important evidence to inform targeted interventions, improve awareness and access to emergency contraception, and ultimately contribute to reducing unintended pregnancies and improving reproductive health outcomes in the region.

## Materials and methods

### Study design and setting

This study was a community-based cross-sectional study that used structured questionnaires for data collection and analysis. The study was conducted in Lira District located in Northern Uganda from 20th June 2023–5th July 2023. Lira District is located in the Lango sub-region of Northern Uganda, approximately 100 kilometers southeast of Gulu City, the largest urban center in the region and about 337 kilometers north of Kampala, the capital city of Uganda. Lira District, like the rest of Northern Uganda, faces significant reproductive health challenges. Notably, the region has one of the highest rates of induced abortion in the country, estimated at 57 per 1,000 women aged 15–49 years [11,12]. Contraceptive use is notably low in this area, with a prevalence of non-use estimated at 55% [10]. Additionally, emergency contraceptive use in Lango sub-region, where Lira District is located, is a mere 0.2%, significantly lower compared to other regions such as South-Central Uganda at 0.4% [7]. The district also has an estimated population of 478,500, with 248,100 females and 230,400 males [13,14]. The majority of residents in Lira District engage in subsistence agriculture, small-scale trading, and informal employment, which are the main sources of livelihood in the area [15]. The population is predominantly composed of the Lango ethnic group. Christianity is the dominant religion, with most residents identifying as Roman Catholic, Anglican, or Pentecostal, while a smaller proportion practice Islam and traditional beliefs. Religious beliefs and cultural norms in the area may influence perceptions and utilization of reproductive health services, including emergency contraception [13,14].

### Study population

Women in the reproductive age of 15–49 years available in the selected sub-counties were included in the study. The target population was women of reproductive age within the selected sub-counties and the accessible population was the women of reproductive age within the selected household.

### Inclusion criteria and exclusion criteria

The study included all women of reproductive age (15–49 years) residing in the selected sub-counties who consented to participate. Women in this age group who reported consistent use of regular, effective contraceptive methods (e.g., injectables, implants, IUDs, or daily pills) with no recent missed doses or lapses in adherence were excluded. Additionally,

women who were unable to hear or speak, as well as those who did not provide consent or assent to participate, were excluded from the study.

## Sample size determination and sampling technique

Kish Leslie (1965) formula was used to estimate the sample size [16] of the study. Where Z = 1.96 at 95% confidence interval, P = 23% which is the prevalence of emergency contraceptive use in Kibera in Nairobi [17], d = 0.05. The resulting sample was adjusted for design effect of 2 and non-response rate of 10% which generated a total sample size of 600 respondents.

$$n = \frac{z^2 P(1-P)}{d^2}$$

$$n = 545 + (545 * 10\%) = 600 \text{ respondents}$$

The study used a multi-stage cluster sampling method. In the first stage three sub-counties were selected using simple random sampling out of five sub-counties in Lira District. In the second stage one parish was selected using simple random sampling in each of the three sub counties which were selected. In each of the parishes 3 villages were selected using simple random sampling. The sampling frame was then determined in each of the villages and the number of participants each village would contribute to the sample was calculated (proportionate by size). Sampling of the respondents was done with the help of the local leaders and the village health team members who are well conversant with the households in their communities. See Table 1 below.

## Data collection method and tools

Data were collected using researcher-administered questionnaires developed by the research team. The questionnaire was initially drafted based on the study objectives and then piloted, reviewed, and refined to improve clarity, relevance, and suitability for the study population. Data collection was conducted by five female research assistants, all of whom were university graduates and familiar with the local context. Prior to fieldwork, the research assistants underwent a two-day training conducted by the research team together with a member of the Research Ethics Committee. The training covered the study objectives, ethical issues, informed consent procedures, interviewing skills, and administration of the questionnaire.

**Table 1. Multi-stage sampling used to select participants of the study.**

| Sub county (Simple random sampling) | Parish (Simple random sampling) | Village (Simple random sampling) | Total number of households | Planned sample (n = 600) | Actual sample (n = 585) |
|---|---|---|---|---|---|
| Adekokwok | Akia | Agali | 143 | 86 | 85 |
| | | Oyea | 114 | 68 | 68 |
| | | Teobwolo | 117 | 70 | 70 |
| Aromo | Bar pii | Otara | 111 | 67 | 66 |
| | | Dag Apele | 112 | 67 | 57 |
| | | Ogot | 109 | 66 | 65 |
| Barr | Ayira | Akutu | 92 | 50 | 49 |
| | | Omara | 114 | 69 | 68 |
| | | Adip | 96 | 57 | 57 |
| | | **Total** | **998** | **600** | **585** |

The tool was pretested among 60 women of reproductive age who met the study inclusion criteria in Ayer Sub-county, Kole District, a setting neighboring Lira District and comparable to the actual study area. This pretesting helped assess the clarity, appropriateness, and effectiveness of the questionnaire in a similar context. The questionnaire achieved a Content Validity Index (CVI) of 80%, which was considered acceptable. Since the tool was a structured questionnaire rather than a psychometric scale, reliability analysis such as internal consistency was not considered applicable.

## Operational definitions

**Emergency contraception:** Emergency contraception was defined as methods of contraception that can be used to prevent pregnancy after unprotected sexual intercourse. These are recommended for use within 5 days but are more effective the sooner they are used after the act of intercourse [4].

**Emergency contraceptive pill use:** Was defined as the use of emergency pills (also called morning after pill) after a previous risky sexual encounter that could result in unintended pregnancy [18]. This study considered the use of any of the most commonly used pills in Uganda including: Postinor-2, Norethisterone, Back-up pill, Microgynon ED Fe and Lydia (Levonorgestrel) within the previous month.

**Women of reproductive age:** Were defined as women in the age group of 15–49 years of age [19].

**Study variables.** Knowledge about emergency contraceptive pill use was measured using eight questions. Each question had three response options, which were scored based on correctness: correct responses were assigned 3 points, partially correct responses were assigned 2 points, and incorrect or "don't know" responses were assigned 1 point. The total knowledge score therefore ranged from 8 to 24. Respondents who scored 13 or above were categorized as having good knowledge, while those scoring 12 or below were categorized as having poor knowledge. The cutoff point was defined by the authors based on the midpoint of the possible score range and was used to distinguish respondents with relatively higher knowledge from those with lower knowledge [20,21].

For attitudes, five questions were asked in the Likert scale with 5 options coded in ascending order: Strongly disagree (1), Disagree (2), Neutral (3), Agree (4), Strongly agree (5). The Responses of all the questions were added for every respondent and a score was generated. The highest score was 25 and the lowest was 5, respondents who scored above 15 were considered to have good attitudes towards use of emergency contraceptive pills and those who scored 15 and below were considered to have poor attitudes towards emergency contraceptive pill use

The outcome variable was the practice of emergency contraceptive pill use and was measured in terms of use of an emergency contraceptive pill in the previous month. The responses were "yes" or "no".

## Data management and analysis

Data were analyzed using SPSS version 26. Descriptive statistics were performed at the univariate level to summarize the frequencies and percentages of key study variables. At the bivariate level, binary logistic regression was used to assess the association between each independent variable and the outcome variable (emergency contraceptive pill use), using a 95% confidence interval. Crude Odds Ratios (COR) and corresponding $p$-values were reported, with a significance threshold set at $p < 0.05$.

Variables with a $p$-value less than 0.20 at the bivariate level were included in the multivariate logistic regression model to control for potential confounders. In this particular study all variables in the sociodemographic and the overall knowledge and attitude were included in the model. In the multivariate analysis, Adjusted Odds Ratios (AOR) and $p$-values were calculated. Variables with $p < 0.05$ in this final model were considered to be significantly associated with emergency contraceptive pill use.

## Ethical considerations

The research protocol was reviewed and approved by the Lira University Research Ethics Committee (LUREC-2023-11). Administrative clearance was also obtained from the office of the District Health officer, and local council chairpersons

of different cells/villages in Lira District. Written informed consent was obtained from the participants and the parents or guardians to the participants below 18 years of age. Assent was obtained from the respondents below 18 years of age. To ensure privacy during questionnaire administration, interviews were conducted in a private setting within or near the participant's home, out of hearing range of other household members. Only the participant and the research assistant were present during the interview, which helped maintain confidentiality and promote open responses on this sensitive topic. The study was conducted in accordance with the Declaration of Helsinki.

## Results

### Demographic characteristics of women of reproductive age 15–49 years in Lira District, Northern Uganda

A total of 585 women of reproductive aged 15–49 years were recruited in this study, resulting in a response rate of 97.5%. A total of 179 (30.6%) of the respondents had used emergency contraception in the previous one month. The mean age of the women was 28.01 years and the standard deviation (SD) was ±7.6 years. Variables that had significant relationship with emergency contraceptive pill use included: age, education level, residence, marital status, income status and source of information about emergency contraceptive pill use. See Table 2 Below.

### Knowledge about emergency contraceptives pill use among women of reproductive age in Lira District, Northern Uganda

Table 3 shows that 63.93% of women were classified as having good knowledge of emergency contraception, while 36.07% had poor knowledge. Most respondents (67.01%) reported that emergency contraceptives prevent pregnancy by preventing implantation, and 63.42% correctly identified common brands such as Postinor and Lydia. Additionally, 64.44% knew that emergency contraceptives should be used within 24–72 hours after unprotected intercourse. Just over half (54.19%) correctly understood that emergency contraception is not an abortion method.

However, important misconceptions were observed. Nearly half (47.69%) believed that high doses of combined oral contraceptives (COCs) and progestin-only pills (POPs) cannot serve as emergency contraceptives, indicating lack of knowledge about available regimens (Yuzpe regimen). Only 18.8% knew that intrauterine devices (IUDs) can be used for emergency contraception, and 41.37% were unaware of this option. Furthermore, 63.76% believed emergency contraceptives are only available in pharmacies, and 52.65% incorrectly thought some are administered by injection, highlighting persistent knowledge gaps.

### Attitudes towards of emergency contraceptive pill use among women of reproductive age in Lira District, Northern Uganda

Nearly half of the participants (47.5%) were neutral on the statement *"I can use emergency contraceptive pills whenever I want to,"* while 29.4% agreed and 15.2% strongly agreed. Similarly, a majority of respondents were neutral regarding the statement *"Emergency contraceptive pills are safe"* (52.8%), and the same proportion (52.8%) were also neutral about the statement *"Emergency contraceptives are effective."* More than half of the respondents (56.9%) expressed neutrality to the statement "I would advise a friend to use emergency contraceptive pills when there is need". Likewise, 58.5% were neutral about the statement *"Emergency contraceptive pills are convenient to use."* Overall, two-thirds of respondents (66.0%) demonstrated poor attitudes towards emergency contraceptive pill use, while only 34.0% had good attitudes. See table 4

### Factors associated with emergency contraceptive pill use among women of reproductive age (15–49) in Lira District, Northern Uganda

In multivariable logistic regression, marital status, knowledge, and attitudes were independently associated with ECP use. Unmarried respondents had higher odds of emergency contraceptive pill use compared to those who were married

**Table 2. Demographic characteristics of women of reproductive age 15-49 years and their relationship with emergency contraceptive pill use in Lira District, Northern Uganda.**

| Variables | Frequency N(%) | Emergency contraceptive pill use | | COR(CI) | P-value |
|---|---|---|---|---|---|
| | | Not used N(%) | Use N(%) | | |
| **Emergency contraceptive pill use** | | **406 (69.40)** | **179 (30.60%)** | | |
| **Age** | | | | | |
| 15-19 | 87(14.87) | 72(82.76) | 15(17.24) | Ref | |
| 20-24 | 106(18.12) | 74(69.81) | 32(30.19) | 2.07(1.03-4.15) | 0.039* |
| 25-29 | 175(29.91) | 118(67.43) | 57(32.57) | 2.31(1.22-4.39) | 0.010* |
| 30-34 | 86(14.70) | 51(59.30) | 35(40.70) | 3.29(1.63-6.65) | 0.001** |
| 35-39 | 71(12.14) | 51(71.83) | 20(28.17) | 1.63(0.88-4.02) | 0.103 |
| 40-49 | 60(10.26) | 40(66.67) | 20(33.33) | 2.4(1.10-5.19) | 0.026* |
| **Employment status** | | | | | |
| Unemployed | 297(50.77) | 199(67) | 98(33) | Ref | |
| employed | 288(49.23) | 207(71.88) | 81(28.13) | 0.79(0.55-1.13) | 0.202 |
| **Education level** | | | | | |
| Primary education | 125(21.37) | 100(80) | 25(20) | Ref | |
| Secondary education | 188(32.14) | 116(61.7) | 72(38.3) | 2.48(1.46-4.20) | 0.001* |
| Tertiary education | 130(22.22) | 66(50.77) | 64(49.23) | 3.87(2.22-6.77) | <0.001*** |
| No formal education | 142(24.27) | 124(87.32) | 18(12.68) | 0.58(0.3-1.12) | 0.107 |
| **Residence** | | | | | |
| Rural | 362(61.88) | 269(74.31) | 93(25.69) | Ref | |
| Urban | 223(38.12) | 137(61.43) | 86(38.57) | 1.82(1.27-2.6) | 0.001** |
| **Marital status** | | | | | |
| Married | 351(60.0) | 257(73.22) | 94(26.78) | Ref | |
| Unmarried | 234(40.0) | 149(63.95) | 84(36.05) | 1.54(1.08-2.20) | 0.017* |
| **Income level** | | | | | |
| Below 100 USD | 293(50.08) | 231(78.84) | 62(21.16) | Ref | |
| 101-200 USD | 75(12.82) | 43(57.33) | 32(42.67) | 2.77(1.62-4.74) | <0.001*** |
| 201-300 USD | 16(2.74) | 8(50) | 8(50) | 3.72(1.34-10.3) | 0.011 |
| 301USD and above | 8(1.37) | 2(25) | 6(75) | 11.17(2.2-56.7) | 0.004 |
| **Source of information** | | | | | |
| School | 117(20.0) | 78(66.67) | 39(33.33) | Ref | |
| Friends | 147(25.13) | 113(76.87) | 34(23.13) | 0.34(0.20-0.57) | <0.001*** |
| Social media | 18(3.08) | 16(88.89) | 2(11.11) | 2.3(0.77-6.88) | 0.134 |
| Television | 12(2.05) | 9(75) | 3(25) | 0.63(0.19-2.11) | 0.458 |
| Radio | 43(7.35) | 29(67.44) | 14(32.56) | 0.53(0.26-1.07) | 0.079 |
| Newspapers | 15(2.56) | 12(80) | 3(20) | 0.59(0.19-1.77) | 0.347 |
| Family Members | 8(1.37) | 3(37.5) | 5(62.5) | 0.53(0.12-2.33) | 0.403 |
| Health Workers | 49(8.37) | 32(65.31) | 17(34.69) | 1(0.514-1.95) | 0.993 |
| **Overall, Knowledge** | | | | | |
| Good Knowledge | 374 | 214(57.22) | 160(42.78) | Ref | |
| Poor Knowledge | 211 | 192(91) | 19(9) | 0.13(0.08-0.22) | <0.001*** |
| **Overall attitudes** | | | | | |
| Poor attitudes | 386(65.98) | | | Ref | |
| Good attitudes | 199(34.02) | | | 5.07(3.47-7.41) | <0.001*** |

*Level of significance at p<0.05, **level of significance at p<0.001, p<0.05

**Table 3. Showing knowledge level of women of reproductive age 15-49 years in Lira District, Northern Uganda.**

| Variable | Frequency (N) | Percentage (%) |
|---|---|---|
| **Emergency contraceptives prevent implantation** | | |
| Yes | 392 | 67.01 |
| No | 16 | 2.74 |
| I don't know | 177 | 30.26 |
| **High doses of COCs and POPs can work as emergency contraceptives** | | |
| No | 279 | 47.69 |
| Yes | 94 | 16.07 |
| I don't know | 212 | 36.24 |
| **Postinor and Lydia are common emergency contraceptives pills** | | |
| Yes | 371 | 63.42 |
| No | 30 | 5.13 |
| I don't know | 184 | 31.45 |
| **When are emergency contraceptives recommended after sex** | | |
| within 24–72 hours | 377 | 64.44 |
| within 24–120 hours | 41 | 7.01 |
| I don't know | 167 | 28.55 |
| **Emergency contraception is an abortion method** | | |
| No | 317 | 54.19 |
| Yes | 60 | 10.26 |
| don't know | 208 | 35.56 |
| **Emergency contraceptives are only found in pharmacies** | | |
| Yes | 373 | 63.76 |
| No | 35 | 5.98 |
| don't Know | 177 | 30.26 |
| **IUDs can be used as Emergency contraceptives** | | |
| Yes | 110 | 18.8 |
| No | 233 | 39.83 |
| I don't know | 242 | 41.37 |
| **Some emergency contraceptives are injected into the body** | | |
| No | 55 | 9.4 |
| Yes | 308 | 52.65 |
| I don't know | 222 | 37.95 |
| **Overall knowledge** | | |
| Good knowledge | 374 | 63.93 |
| Poor Knowledge | 211 | 36.07 |

(AOR = 2.85, 95% CI 1.83–4.43, p < 0.001). Respondents who had poor knowledge about emergency contraceptive pills had lower odds of emergency contraceptive pill use compared to those who had good knowledge (AOR = 0.11,95%CI 0.07–0.21, p < 0.001). Finally, respondents who had good attitudes towards use of emergency contraceptive had higher odds of emergency contraceptive pill use compared to those who had poor attitudes (AOR = 3.89, 95%CI 2.59–5.84, p < 0.001). See Table 5

**Table 4. Showing the attitudes of women of reproductive age 15-49 years in Lira District, Northern Uganda.**

| Variable | Frequency (N) | Percentage (%) |
|---|---|---|
| **I can use emergency contraceptive pill whenever I want to** | | |
| Strongly Disagree | 26 | 4.44 |
| Disagree | 20 | 3.42 |
| Neutral | 278 | 47.52 |
| Agree | 172 | 29.4 |
| Strongly Agree | 89 | 15.21 |
| **Emergency contraceptive pills are safe** | | |
| Strongly Disagree | 24 | 4.1 |
| Disagree | 31 | 5.3 |
| Neutral | 309 | 52.82 |
| Agree | 161 | 27.52 |
| Strongly Agree | 60 | 10.26 |
| **Emergency contraceptives are effective** | | |
| Strongly Disagree | 28 | 4.79 |
| Disagree | 33 | 5.64 |
| Neutral | 332 | 56.75 |
| Agree | 134 | 22.91 |
| Strongly Agree | 58 | 9.91 |
| **I would advise a friend to use emergency contraceptive pills when there is need** | | |
| Strongly Disagree | 32 | 5.47 |
| Disagree | 38 | 6.5 |
| Neutral | 333 | 56.92 |
| Agree | 138 | 23.59 |
| Strongly Agree | 44 | 7.52 |
| **Emergency contraceptive pills are convenient to use** | | |
| Strongly Disagree | 30 | 5.13 |
| Disagree | 32 | 5.47 |
| Neutral | 342 | 58.46 |
| Agree | 125 | 21.37 |
| Strongly Agree | 56 | 9.57 |
| **Overall attitudes** | | |
| Poor attitudes | 386 | 65.98 |
| Good attitudes | 199 | 34.02 |

## Discussion

This study assessed the knowledge, attitudes, and practices (KAP) regarding emergency contraceptive pill (ECP) use among women aged 15–49 years in Lira District, and identified factors associated with use. The results of the study show that emergency contraceptive pill use was modest: 179 (30.6%) reported using ECPs in the previous month. knowledge was generally good (63.94% had good knowledge), attitudes were largely negative (65.98%), suggesting that awareness does not necessarily translate into acceptance. The factors associated include unmarried status, good knowledge, and positive attitudes (P<0.05).

**Table 5. Showing factors associated with emergency contraceptive pill use among women of reproductive age 15-49 years in Lira District, Northern Uganda.**

| Variable | COR (CI) | P value | AOR (CI) | P value |
|---|---|---|---|---|
| **Marital status** | | | | |
| Married | Ref | | Ref | |
| Unmarried | 1.54(1.08-2.20) | 0.017* | 2.85 (1.83-4.43) | <0.001*** |
| **Overall knowledge** | | | | |
| Good Knowledge | Ref | | Ref | |
| Poor knowledge | 0.13(0.08-0.22) | <0.001*** | 0.11 (0.07-0.21) | <0.001*** |
| **Overall attitudes** | | | | |
| Poor attitude | Ref | | Ref | |
| Good attitude | 5.07(3.47-7.41) | <0.001*** | 3.89 (2.59-5.84) | <0.001*** |

*Level of significance at p<0.05, **level of significance at p<0.001, p<0.05

In this study, 179 (30.6%) respondents reported using emergency contraceptive pills in the previous month, indicating a modest level of utilization among women of reproductive age in Lira District. This finding is slightly higher than the 23% reported by Mutie et al. among women in Kibera, Kenya [17]. The difference may be explained by variations in study period and setting, as the Kibera study was conducted in 2012, when access to reproductive health information and services may have been more limited. However, our finding is lower than the 88% reported among university students in Uganda [22], which may reflect differences in study population, since university students are more likely to have greater exposure to sexual and reproductive health information and services. It is also lower than the 64% reported in Tororo District [23], although that study assessed ever use of emergency contraceptive pills, whereas the present study measured use within the previous month. However, since emergency contraceptive pills are intended for occasional use after unprotected intercourse or contraceptive failure, measuring recent use provides a more specific estimate of current utilization than lifetime use. In contrast, ever-use measures may reflect past exposure rather than current practice. This distinction is important when interpreting findings and making recommendations, since utilization reflects actual recent behavior, whereas access refers to the ability to obtain emergency contraceptives when needed. Therefore, low utilization should not automatically be interpreted as poor access, although the two may be related.

This study also reveals that 374 (63.94%) respondents had good knowledge of emergency contraceptive pill use. This finding is higher than the 27.8% reported by Babatunde et al. in Nigeria [24], but is comparable to the 63% reported in a study from India [25]. These differences may be due to variation in access to information, education level, and socio-cultural context across study settings [26,27]. In our study, many respondents reported obtaining information from schools and friends, which may partly explain the relatively high level of knowledge observed. Knowledge was also significantly associated with emergency contraceptive pill utilization. Women with poor knowledge were 89% less likely to use emergency contraceptive pills than those with good knowledge (AOR=0.11, 95% CI: 0.07–0.21, p<0.001). This is consistent with findings from Ethiopia, where better knowledge was associated with greater use of emergency contraception [28]. This suggests that improving women's understanding of emergency contraceptive pills, including their timing, method options, and purpose, may support appropriate utilization.

Most respondents, 386 (65.98%), reported negative attitudes toward emergency contraceptive pill use. This may reflect persistent misconceptions about emergency contraception within the community. Our findings differ from those of a study in Nepal, which reported that 93.4% of participants had positive attitudes toward emergency contraception [29]. This difference may be explained by variations in study setting and access to information. Shakya et al. conducted their study in a more urban area, where exposure to reproductive health information and services is typically higher. In contrast, our study was conducted in Northern Uganda, a resource-constrained setting where limited access to accurate information

may contribute to negative attitudes. This finding is important for policymakers because it justifies community engagement strategies involving VHTs, women leaders, and youth champions to shift norms and improve acceptability.

Respondents with positive attitudes towards emergency contraceptive pill had 3.89 higher odds of Emergency contraceptive use pill compared to those with poor attitudes (AOR = 3.89, 95% CI 2.59–5.84, p < 0.001). This result aligns with studies showing that a positive attitude towards contraception increases the likelihood of utilizing that particular contraceptive method [25,30,31]. This may be because a positive attitude enhances a person's intention to engage in a specific behaviour, as illustrated in the Health Belief Model [32].

Our study demonstrates a significant association between marital status and emergency contraceptive pill use. Unmarried women had 2.85 higher odds of emergency contraceptive pill use compared to married women (AOR = 2.85; 95% CI: 1.83–4.43; $p < 0.001$). This may be attributed to the fact that unmarried women, while often sexually active, may not be ready to initiate childbearing and may avoid the routine use of regular contraceptive methods due to social stigma, irregular sexual activity, or limited access [33,34]. As a result, they are more likely to resort to emergency contraception following unprotected intercourse. On the other hand, married women are more likely to adopt regular contraceptive methods as part of ongoing family planning and therefore may not need to use emergency contraceptive pills [35]. These findings underscore the need for policies that tailor sexual and reproductive health interventions to the distinct needs of both unmarried and married women. Efforts should ensure timely, equitable, and stigma-free access to emergency contraceptives to prevent unintended pregnancies.

## Conclusion and recommendations

Emergency contraceptive pill utilization among women of reproductive age in Lira District was modest, despite generally good knowledge. Negative attitudes toward emergency contraceptive pill use were common, and utilization was significantly associated with knowledge, attitudes, and marital status. These findings indicate that improving utilization requires more than awareness alone; it also requires addressing misconceptions, strengthening positive attitudes, and ensuring that women can obtain emergency contraceptive pills when needed.

## Study limitations and strength

This study has several limitations that should be considered when interpreting the findings. The cross-sectional design limits our ability to establish causality; the observed associations between knowledge, attitudes, marital status, and emergency contraceptive pill (ECP) use reflect correlations at one point in time rather than cause–effect relationships. Moreover, ECP use and other key variables were collected using self-reported responses, which are prone to recall bias, especially when participants were asked to remember events such as ECP use in the previous month. Furthermore, emergency contraception is a sensitive topic, responses may also have been affected by social desirability bias, leading some respondents to underreport use or provide answers they perceived as acceptable. Finally, although multi-stage cluster sampling improved coverage, the findings may not be fully generalizable beyond the study setting due to contextual differences in access to information and services. Nonetheless, the findings provide valuable baseline data that can inform future longitudinal and interventional studies aimed at enhancing the understanding and utilization of emergency contraception appropriately, particularly among women in low- and middle-income countries such as Uganda.

## Supporting information

**S1 Data. Final data set. ZIP archive containing the final dataset used in the analyses reported in this manuscript.** (ZIP)

## Acknowledgments

The authors would like to acknowledge the study participants for participating in the study. A special thanks to the faculty of public health Lira university for supporting the research processes.

## Author contributions

**Conceptualization:** Marvin Musinguzi, Josephine Vanessa Nakalema, Amir Kabunga.

**Data curation:** Deo Benyumiza, Joshua Oryem Opido.

**Formal analysis:** Marvin Musinguzi, Eustes Kigongo, Edward Kumakech, Marc Sam Opollo.

**Funding acquisition:** Edward Kumakech.

**Investigation:** Marvin Musinguzi, Josephine Vanessa Nakalema, Deo Benyumiza, Edward Kumakech, Rashida Namuwaya, Joshua Oryem Opido, Nicholas Damulira, Raymond Tumwesigye, Marc Sam Opollo.

**Methodology:** Marvin Musinguzi, Josephine Vanessa Nakalema, Deo Benyumiza, Edward Kumakech, Joshua Oryem Opido, Nicholas Damulira, Raymond Tumwesigye, Amir Kabunga, Marc Sam Opollo.

**Project administration:** Josephine Vanessa Nakalema, Nicholas Damulira, Marc Sam Opollo.

**Resources:** Joshua Oryem Opido.

**Software:** Joshua Oryem Opido, Nicholas Damulira, Raymond Tumwesigye, Marc Sam Opollo.

**Supervision:** Marvin Musinguzi, Eustes Kigongo, Marc Sam Opollo.

**Validation:** Josephine Vanessa Nakalema, Eustes Kigongo, Edward Kumakech, Rashida Namuwaya, Joshua Oryem Opido, Raymond Tumwesigye, Amir Kabunga.

**Visualization:** Marvin Musinguzi, Eustes Kigongo, Amir Kabunga.

**Writing – original draft:** Marvin Musinguzi, Josephine Vanessa Nakalema, Amir Kabunga.

**Writing – review & editing:** Josephine Vanessa Nakalema, Deo Benyumiza, Eustes Kigongo, Edward Kumakech, Rashida Namuwaya, Raymond Tumwesigye, Amir Kabunga, Marc Sam Opollo.

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
