## [Decision Letter · Decision Letter 0]

10 Dec 2024

Dear Dr. Musinguzi,

Thank you for submitting your manuscript to PLOS ONE. After careful consideration, we feel that it has merit but does not fully meet PLOS ONE’s publication criteria as it currently stands. Therefore, we invite you to submit a revised version of the manuscript that addresses the points raised during the review process.

Please submit your revised manuscript  by Jan 24 2025 11:59PM. If you will need more time than this to complete your revisions, please reply to this message or contact the journal office at plosone@plos.org . . A rebuttal letter that responds to each point raised by the academic editor and reviewer(s). You should upload this letter as a separate file labeled 'Response to Reviewers'.A marked-up copy of your manuscript that highlights changes made to the original version. You should upload this as a separate file labeled 'Revised Manuscript with Track Changes'.An unmarked version of your revised paper without tracked changes. You should upload this as a separate file labeled 'Manuscript'.

We look forward to receiving your revised manuscript.

Kind regards,

Yitagesu Habtu Aweke, Ph.D

Academic Editor

PLOS ONE

Journal requirements: When submitting your revision, we need you to address these additional requirements. 1. Please ensure that your manuscript meets PLOS ONE's style requirements, including those for file naming. The PLOS ONE style templates can be found at https://journals.plos.org/plosone/s/file?id=wjVg/PLOSOne_formatting_sample_main_body.pdf and https://journals.plos.org/plosone/s/file?id=ba62/PLOSOne_formatting_sample_title_authors_affiliations.pdf

Additional Editor Comment:

Two expert reviewers have acknowledged that this study is interesting. However, although this topic is important, the expert review team has highlighted several points. Based on their comments and my own reading, I believe the paper has potential but requires significant revisions. Therefore, I invite you to address the reviewers' comments.  Too mention as an example:

Does your finding support your conclusion “Efforts to increase contraceptive access should also enhance information about emergency contraception?”It is important to organize the the result sections such as "descriptive analysis", and " multivariable analysis" in another section, and the tables can be organized together.What does it mean “… cross-sectional study that used quantitative methods for data collection and analysis.”?for data collection and analysis.”?

Reviewers' comments:

Reviewer's Responses to Questions

**Comments to the Author**

1. Is the manuscript technically sound, and do the data support the conclusions?

Reviewer #1: Partly

Reviewer #2: Partly

2. Has the statistical analysis been performed appropriately and rigorously?

Reviewer #1: Yes

Reviewer #2: I Don't Know

3. Have the authors made all data underlying the findings in their manuscript fully available?

Reviewer #1: Yes

Reviewer #2: Yes

4. Is the manuscript presented in an intelligible fashion and written in standard English?

Reviewer #1: No

Reviewer #2: No

Reviewer #1: Line 37 and 106 are contradictory. Authors need to clarify and state which of the dates is correct

Line 38, 130,133,142,197 and 198 should be reviewed by the authors. Authors indicated that they we they going to recruit 600(indeed line 38 says that 600 were recruited), yet using the multi stage sampling to select the participants, 585 were selected and interviewed yet the authors still mentions a response rate of 97.5% and in another breath indicates that all the participants contacted agreed to the interviewed. Authors need to strongly look at this again.

Line 52: Authors should be specific and indicate what component of the marital status, attitude or the knowledge that influences the use of the EC. eg is it the unmarried or the married???

Line 91-92,111:Can the authors report/describe contraceptive use (prevalence) instead of contraceptive non-use as stated in this paper

Line 94-95:why do you select HIV population? Better to give the prevalence in the general population first. If that is not known, then the authors should say so.

Line 114- Much useful information with respect to this study should rather be women in reproductive age and not all women as stated by the authors. They should have a look at it again if the information exist.

Line 122: Did the authors include women who were already on regular effective contraception? that will obviously give different results which should be interpreted differently. Did we include those who were on regular contraception but had missed/delayed the next dose for which they could be at risk of unwanted pregnancy and hence would be candidate for EC? Authors should clarify these because they have implications on the findings. They can not be lumped together. Again if a person is looking for pregnancy then obvious EC does not come in all. The authors should be much clearer on the eligibility criteria.

Line 146: Where did you pre-test the tool??

Line 155- suggested to authors that it should be 'after unprotected sexual intercourse'

Line 167-167: frame this statement properly..... it looks like an important conjunction 'and' is missing somewhere and makes the it hanging

Line 200: kindly refer to the study participants as women and not girls

Line 253:what was the duration for which the use of EC was studied (was it the previous one month as the present study or different?)

Line 255: what was the setting here, these should come out......was this also in Uganda or the northern part of Uganda as well??

Line 259:was this also assessed over the previous month? what was the inclusion and exclusion criteria; it is likely to affect this in addition that this was not general community based population

Line 273: was this study also in Uganda? and could the study population and educational levels accounted for it???

Line 274: your discussion around this point is very unclear....what exactly do authors want to convey??

Line 280:to what information, what platform exist in Uganda that promote accessibility to this information, authors need to bring this out

Line 282-288:repeation of paragraph 270-276 to this section ..........it is repetition of the same information

Line 320:which study was authors referring to? Recommending that the authors report on the figures they are referring to.

Line 325:causality cannot be determined in cross sectional study.....it may be associated but not causality as the authors put it. They should revise it.

Line 332: Incomplete sentence. Authors should address it.

Line 382-385: The reference is captured with block letters and it is not consistent with what the other references have been captured.

Table 2: With respect to age, Which category did the authors put the 18 year olds? and those who were 24 and 35 years....need to re categorize properly. what informs this sub groupings, there appears to be no science behind it......could refer to the groupings used by your DHS or some better sub categorization.

Authors should have a second look at the following listed paragraphs and address the applicable grammatical errors, typos, concordance of sentences, repetition of words, appropriate use of capital and small letters and fragmentation of words among others...55, 62, 107, 117,118,122,128,129,140,141,158,170,171,184,202,205,232,233,242,244,295,299,301 and 321.

Reviewer #2: The manuscript addresses an important topic, but several sections require significant improvement in terms of clarity, consistency, and scientific rigor. Below is detailed feedback on specific sections and issues:

Background:

1. Clarity and Grammar:

o Line 55: The sentence "Contraceptive use is a challenge globally are a public health challenge" is grammatically incorrect and redundant. Please revise for clarity.

o There is inconsistency in quantitative data related to the number of women of reproductive age and the need for family planning. For example, figures on contraceptive use in Uganda are unclear (350 vs. 40). These data points need to be clarified or synthesized for coherence.

2. Redundancy in References:

o Reference [2] is repeated unnecessarily. It should be cited only once, ideally at the end of the relevant statement.

3. Transitions and Word Choice:

o Line 60: The use of “However” is not contextually appropriate. Please revise the sentence to better reflect the logical flow.

4. Structure and Writing Style:

o The background section would benefit from a thorough review by a native English speaker to improve clarity, formal tone, and scientific rigor.

o Some paragraphs are verbose and could be synthesized into concise, scientifically written statements. This comment can be directed to the whole paper.

Methods:

1. Variable Selection for Multivariate Analysis:

o Authors need to describe the process used for selecting variables included in the multivariate model. It is unclear if the variables shown in the tables are the only ones included or if others were considered but excluded. Authors need to specify the process by which they selected the variables for the multivariable analysis.

2. Specific Variables:

o The variable religion is mentioned in the methods but not presented in the results. Please include or justify its omission.

o Consider discussing other relevant variables, such as maternal education and sexual behaviors, which could add depth to the analysis.

3. Time Frame Definition:

o The study focuses on contraceptive use in the last month. If women who used emergency contraception more than once but not in the past month were categorized as non-users, this needs to be clarified in the objectives. Distinguish between factors associated with general use versus use in the previous month.

Results:

1. Table Presentation:

o The title of Table 5 appears misaligned or misplaced. Please correct formatting.

o In Table 2, the section on contraceptive pill use is unclear. Specify whether this refers to contraceptive pills in general or emergency contraceptive pills. If it is emergency contraception, ensure consistency in column headings.

o I recommend presenting the results related to knowledge and attitudes before discussing the factors associated with the outcome. Specifically, the univariate and multivariate analyses should follow this order for better flow and clarity

Discussion:

1. Relevance and Precision:

o The discussion claims that individual factors, knowledge, and attitudes are associated with the outcome, but only marital status is included in the multivariate analysis. This discrepancy needs to be addressed.

2. Comparative Analysis:

o The discussion compares prevalence rates with other studies but does not address whether the definition of emergency contraceptive use (e.g., limited to the last month) aligns across studies. This difference could significantly impact comparisons and must be discussed.

3. Causal Inferences:

o The authors mention the importance of access to information but do not provide data supporting this claim. Conclusions should be based only on findings from the study and not unsupported assumptions.

4. General Writing:

o The discussion requires polishing for better flow and coherence. Revise to align statements with the presented results and ensure logical arguments.

Final Notes:

1. Consistency in Tables:

o Ensure all tables are correctly labeled and formatted. Double-check the alignment of titles and contents.

2. Language and Formatting:

o The manuscript needs a detailed language review to improve grammar, syntax, and adherence to scientific standards

**Do you want your identity to be public for this peer review?** For information about this choice, including consent withdrawal, please see our For information about this choice, including consent withdrawal, please see our Privacy Policy .

Reviewer #1: No

Reviewer #2: No

While revising your submission, please upload your figure files to the Preflight Analysis and Conversion Engine (PACE) digital diagnostic tool, https://pacev2.apexcovantage.com/ . PACE helps ensure that figures meet PLOS requirements. To use PACE, you must first register as a user. Registration is free. Then, login and navigate to the UPLOAD tab, where you will find detailed instructions on how to use the tool. If you encounter any issues or have any questions when using PACE, please email PLOS at . PACE helps ensure that figures meet PLOS requirements. To use PACE, you must first register as a user. Registration is free. Then, login and navigate to the UPLOAD tab, where you will find detailed instructions on how to use the tool. If you encounter any issues or have any questions when using PACE, please email PLOS at figures@plos.org . Please note that Supporting Information files do not need this step.. Please note that Supporting Information files do not need this step.

---

## [Author Response · Author response to Decision Letter 1]

5 Jun 2025

Reviewer #1:

Line 37 and 106 are contradictory. Authors need to clarify and state which of the dates is correct

Thank you for this observation the authors have corrected this contradicting statement

Line 38, 130,133,142,197 and 198 should be reviewed by the authors. Authors indicated that they we they going to recruit 600(indeed line 38 says that 600 were recruited), yet using the multi stage sampling to select the participants, 585 were selected and interviewed yet the authors still mentions a response rate of 97.5% and in another breath indicates that all the participants contacted agreed to the interviewed. Authors need to strongly look at this again. We apologise for confusion the 600 was the estimated sample size but 585 were recruited for the study

Line 52: Authors should be specific and indicate what component of the marital status, attitude or the knowledge that influences the use of the EC. eg is it the unmarried or the married??? This has been specified to the un married

Line 91-92,111:Can the authors report/describe contraceptive use (prevalence) instead of contraceptive non-use as stated in this paper That you for this observation, this has been corrected as advised

Line 94-95: why do you select HIV population? Better to give the prevalence in the general population first. If that is not known, then the authors should say so. This was the only study we came across in lira district that has information about unwanted pregnancy. Never the less we decided to remove this and noted it is unknown

Line 114- Much useful information with respect to this study should rather be women in reproductive age and not all women as stated by the authors. They should have a look at it again if the information exist. This has been adjusted as advised. Thank You

Line 122: Did the authors include women who were already on regular effective contraception? that will obviously give different results which should be interpreted differently. Did we include those who were on regular contraception but had missed/delayed the next dose for which they could be at risk of unwanted pregnancy and hence would be candidate for EC? Authors should clarify these because they have implications on the findings. They can not be lumped together. Again, if a person is looking for pregnancy, then obvious EC does not come in all. The authors should be much clearer on the eligibility criteria. Thank you for this insightful observation. We agree that distinguishing between women on regular contraception and those not using it—or using it irregularly is crucial in interpreting EC use data accurately. We have now clarified our eligibility criteria and explained how we addressed this issue in our study design and analysis.

Line 146: Where did you pre-test the tool?? The tool was pretested in Kole district and this information has been added to the data collection tools

Line 155- suggested to authors that it should be 'after unprotected sexual intercourse' Thank you for this suggestion this had been included in the write up

Line 167-167: frame this statement properly..... it looks like an important conjunction 'and' is missing somewhere and makes the it hanging This has been addressed thank you

Line 200: kindly refer to the study participants as women and not girls Thank you for this observation this has been addressed

Line 253: what was the duration for which the use of EC was studied (was it the previous one month as the present study or different?) Thank you for this clarification EC use was assed for the previous month

Line 255: what was the setting here, these should come out......was this also in Uganda or the northern part of Uganda as well?? Thank you for this observation this has been addressed

Line 259: was this also assessed over the previous month? what was the inclusion and exclusion criteria; it is likely to affect this in addition that this was not general community-based population This was not assed over the period of one month

Line 273: was this study also in Uganda? and could the study population and educational levels accounted for it???

Thank you for the observation. This has been clarified in the write up

Line 274: your discussion around this point is very unclear....what exactly do authors want to convey??

Thank you for this observation the paragraph has been re written to clarify the issue

Line 280:to what information, what platform exist in Uganda that promote accessibility to this information, authors need to bring this out Thank you for this concerned this has been addressed

Line 282-288:repeation of paragraph 270-276 to this section ..........it is repetition of the same information

Thank you for this observation this has been removed from the write up

Line 320: which study was authors referring to? Recommending that the authors report on the figures they are referring to. Thank you for this observation. this has been corrected

Line 325: causality cannot be determined in cross sectional study....it may be associated but not causality as the authors put it. They should revise it. This has been revised as advised thank you.

Line 332: Incomplete sentence. Authors should address it.

Thank you for this

Line 382-385: The reference is captured with block letters and it is not consistent with what the other references have been captured. Tis has been corrected thank you

Table 2: With respect to age, which category did the authors put the 18-year-olds? and those who were 24 and 35 years.... need to re categorize properly. what informs this sub groupings, there appears to be no science behind it......could refer to the groupings used by your DHS or some better sub categorization.

We thank the reviewer for this important observation. We acknowledge that the initial age groupings lacked explicit justification and may have created ambiguity around boundary ages. We have now revised the age categories in accordance with the Uganda Demographic and Health Survey (UDHS) 2016 classification, which is widely recognized in reproductive health research. Specifically, we reclassified age into the following categories: 15–19, 20–24, 25–29, 30–34, 35–39, and 40–49 years. Each participant was assigned to a group based on completed age. This revision ensures comparability with national data and enhances the scientific rigor of our analysis. Table 2 and related text have been updated accordingly.

Authors should have a second look at the following listed paragraphs and address the applicable grammatical errors, typos, concordance of sentences, repetition of words, appropriate use of capital and small letters and fragmentation of words among others...55, 62, 107, 117,118,122,128,129,140,141,158,170,171,184,202,205,232,233,242,244,295,299,301 and 321. Thank you for this observation the manuscript has been revised for all typos and errors

Reviewer 2

Background:

1. Clarity and Grammar:

o Line 55: The sentence "Contraceptive use is a challenge globally are a public health challenge" is grammatically incorrect and redundant. Please revise for clarity. Thank you for this observation this has been addressed

o There is inconsistency in quantitative data related to the number of women of reproductive age and the need for family planning. For example, figures on contraceptive use in Uganda are unclear (350 vs. 40). These data points need to be clarified or synthesized for coherence.

2. Redundancy in References:

o Reference [2] is repeated unnecessarily. It should be cited only once, ideally at the end of the relevant statement.

This has been corrected

3. Transitions and Word Choice:

o Line 60: The use of “However” is not contextually appropriate. Please revise the sentence to better reflect the logical flow. Thank you for this observation the word However has been removed from the sentence

4. Structure and Writing Style:

o The background section would benefit from a thorough review by a native English speaker to improve clarity, formal tone, and scientific rigor.

This has been reviewed by the authors. Thank you

o Some paragraphs are verbose and could be synthesized into concise, scientifically written statements. This comment can be directed to the whole paper.

The manuscript was rewritten to address this comment

Methods:

1. Variable Selection for Multivariate Analysis:

o Authors need to describe the process used for selecting variables included in the multivariate model. It is unclear if the variables shown in the tables are the only ones included or if others were considered but excluded. Authors need to specify the process by which they selected the variables for the multivariable analysis. The process by which variables were selected for the multivariate analysis has been added to the manuscript in the data analysis section

2. Specific Variables:

o The variable religion is mentioned in the methods but not presented in the results. Please include or justify its omission. Although religion was initially considered during the development of the questionnaire, it was excluded from the final tool to maintain brevity and focus on variables with stronger theoretical relevance to emergency contraceptive use

o Consider discussing other relevant variables, such as maternal education and sexual behaviors, which could add depth to the analysis. With respect to maternal education, our study captured the participants' level of education, which is already presented in the results (Table 2) and discussed as a factor associated with emergency contraceptive use.

We acknowledge that sexual behavior variables, such as the number of recent sexual partners or condom use, were not collected in this study. We recognize this as a limitation and have included a statement in the discussion to reflect this. We agree that incorporating such variables in future research would provide a more comprehensive understanding of emergency contraceptive use.

3. Time Frame Definition:

o The study focuses on contraceptive use in the last month. If women who used emergency contraception more than once but not in the past month were categorized as non-users, this needs to be clarified in the objectives. Distinguish between factors associated with general use versus use in the previous month.

Thank you for this observation

Results:

1. Table Presentation:

o The title of Table 5 appears misaligned or misplaced. Please correct formatting. Thank you for this observation the title has been corrected

o In Table 2, the section on contraceptive pill use is unclear. Specify whether this refers to contraceptive pills in general or emergency contraceptive pills. If it is emergency contraception, ensure consistency in column headings. This has been addressed in table 2 thank you

o I recommend presenting the results related to knowledge and attitudes before discussing the factors associated with the outcome. Specifically, the univariate and multivariate analyses should follow this order for better flow and clarity The order of presenting the results is as follows:

1. Sociodemographic variables

2. Knowledge towards use of emergency contraceptive pill

3. attitude towards use of emergency contraceptive pill

4. factors associated with emergency contraceptive use

Discussion:

1. Relevance and Precision:

o The discussion claims that individual factors, knowledge, and attitudes are associated with the outcome, but only marital status is included in the multivariate analysis. This discrepancy needs to be addressed.

Overall knowledge and overall attitudes are the ones associated with emergency contraceptive use.

These are composite variables

2. Comparative Analysis:

o The discussion compares prevalence rates with other studies but does not address whether the definition of emergency contraceptive use (e.g., limited to the last month) aligns across studies. This difference could significantly impact comparisons and must be discussed.

Thank you for this observation

3. Causal Inferences:

o The authors mention the importance of access to information but do not provide data supporting this claim. Conclusions should be based only on findings from the study and not unsupported assumptions. We acknowledge that our study defined emergency contraceptive pill use as use within the previous one month, whereas some of the studies we compared our findings with assessed lifetime or ever use. This difference in definitions could contribute to discrepancies in reported prevalence rates. We have now revised the discussion section to clearly highlight this distinction and caution against direct comparisons where timeframes differ.

4. General Writing:

o The discussion requires polishing for better flow and coherence. Revise to align statements with the presented results and ensure logical arguments. The discussion sections have been revised ad advised thank you

Final Notes:

1. Consistency in Tables:

o Ensure all tables are correctly labeled and formatted. Double-check the alignment of titles and contents. This has been addressed as advised thank you

2. Language and Formatting:

o The manuscript needs a detailed language review to improve grammar, syntax, and adherence to scientific standards We reviewed the language in the manuscript as advised. Thank you

---

## [Decision Letter · Decision Letter 1]

21 Jul 2025

Dear Dr. Marvin Musinguzi ,

Thank you for submitting your manuscript to PLOS ONE. After careful consideration, we feel that it has merit but does not fully meet PLOS ONE’s publication criteria as it currently stands. Therefore, we invite you to submit a revised version of the manuscript that addresses the points raised during the review process.

We look forward to receiving your revised manuscript.

Kind regards,

Yitagesu Habtu Aweke, Ph.D

Academic Editor

PLOS ONE

Journal Requirements:

Reviewers' comments:

Reviewer's Responses to Questions

**Comments to the Author**

Reviewer #2: All comments have been addressed

2. Is the manuscript technically sound, and do the data support the conclusions?

Reviewer #2: No

3. Has the statistical analysis been performed appropriately and rigorously?

Reviewer #2: I Don't Know

4. Have the authors made all data underlying the findings in their manuscript fully available?

Reviewer #2: Yes

5. Is the manuscript presented in an intelligible fashion and written in standard English?

Reviewer #2: No

Reviewer #2: (No Response)

**Do you want your identity to be public for this peer review?** For information about this choice, including consent withdrawal, please see our For information about this choice, including consent withdrawal, please see our Privacy Policy .

Reviewer #2: No

While revising your submission, please upload your figure files to the Preflight Analysis and Conversion Engine (PACE) digital diagnostic tool, https://pacev2.apexcovantage.com/ . PACE helps ensure that figures meet PLOS requirements. To use PACE, you must first register as a user. Registration is free. Then, login and navigate to the UPLOAD tab, where you will find detailed instructions on how to use the tool. If you encounter any issues or have any questions when using PACE, please email PLOS at . PACE helps ensure that figures meet PLOS requirements. To use PACE, you must first register as a user. Registration is free. Then, login and navigate to the UPLOAD tab, where you will find detailed instructions on how to use the tool. If you encounter any issues or have any questions when using PACE, please email PLOS at figures@plos.org . Please note that Supporting Information files do not need this step.. Please note that Supporting Information files do not need this step.

---

## [Author Response · Author response to Decision Letter 2]

18 Sep 2025

Comments Responses

1. Clarity and Logic of Study Design

The inclusion criteria should be described before the exclusion criteria for clarity and

proper flow of the methodology section This has been rewritten as advised thank you

The secondary objective—identifying factors associated with emergency contraceptive

use—is not clearly stated in the objectives. This should be made explicit in the

introduction and reflected throughout the manuscript.

The researchers aimed to ascertain whether knowledge and attitudes were associated with practices. To achieve this, both bivariate and multivariate analyses were conducted. However, the main objective of the study was to assess the knowledge, attitudes, and practices related to emergency contraceptive use. Putting it in the title and the main objective of the study is misleading we didn’t comprehensively collect data on other factors that would better achieve this objective

2. Measurement and Cut-Off Clarity

o In line 167, the manuscript mentions that scores above 13 represent "good knowledge." However, this is confusing, as the phrasing implies that a score of 13 is neither good nor poor. Please clarify the cutoff and specify the direction of the scale (i.e., does a higher score indicate better knowledge or attitude?). This concern has been addressed by clarifying that a score of 13 and above was used as the cutoff for good knowledge. Since correct answers were coded with higher values, achieving a higher total score directly reflects a higher level of knowledge among respondents

o Additionally, it is not clear whether this cutoff was validated or arbitrarily chosen by the authors. If it is author-defined, this should be clearly acknowledged with justification. This has been addressed and justification for the cut off have been added

3. Sample Size and Response Rate

o Lines 199–200 state that the calculated sample size was 600, but only 585 participants were recruited. However, this does not constitute a response rate. A response rate refers to the number of respondents relative to the number invited or contacted. If only 585 were contacted, the study did not reach its intended sample size, and this should be transparently described.

Thank you for this observation. It is true that the study did not reach the intended sample size. This has also been transparently described in the study. And therefore the response rate of the study was not 100% but ruther 97.5%

4. Results Structure and Flow

o The structure of the results section should better align with the stated objectives. First, describe the characteristics of the sample, followed by knowledge, attitudes, and emergency contraceptive use. Only then should results pertaining to associated factors (i.e., the secondary objective) be presented.

We appreciate the reviewer’s suggestion regarding restructuring the results section. However, in our study, practice was assessed using only one question about previous use of emergency contraceptive pills in the last 30 days. Given this, we felt it was more appropriate to report practice as a percentage in the text before the first table, rather than creating a separate section.

5. Variables and Analyses

o In the "Study Variables" section (line 163), please explicitly list the variables considered for the multivariate analysis.

o It is unclear which covariates were included in the final model. Please specify these in the results section when presenting the multivariable analysis.

All the variables in the socio demographics and the overall knowledge and attitude were included in the multivariate analysis. This has instead been added to the analysis section of the study

6. Terminology and Consistency

o Ensure consistent use of terms throughout the manuscript. For instance, the methods section uses “good knowledge vs poor knowledge,” but the results refer to “knowledgeable vs not knowledgeable.” Consistency is essential, particularly if the categories are author-defined.

o Spelling and typographical errors should be corrected, such as "un married" (line 241) which should be "unmarried," and the double period on line 233. Thank you for this observation. The manuscript has been revised to only include good knowledge and poor knowledge

7. Table and Figure Presentation

o Table titles should appear directly above the corresponding tables. For example, Table 5

is mentioned in the text before it appears, creating confusion.

Thank you for this observation. it has been addressed as advised

o Table 4 is labeled as referring to knowledge, but the accompanying text addresses

attitudes. This inconsistency should be corrected. This has been addressed thank you

8. Interpretation of Attitudinal Data

o Line 230 refers to neutrality regarding the effectiveness of emergency contraceptives. It is unclear whether this is measuring belief in efficacy or safety. The phrasing should be revised to clarify the item participants responded to (e.g., “The majority of respondents were neutral in response to the statement: ‘Emergency contraceptives are safe and effective.’”).

Thank you for this observation the whole paragraph 230-238 has been rewritten as advised

9. Discussion and Framing

o The discussion begins by emphasizing usage and associated factors, but the primary objective is to assess knowledge, attitudes, and use. Please realign the discussion to reflect the stated goals.

This has been addressed as advised and factors associated has been removed as the primary objective

o The phrase “individual factors” (line 254) is used in the plural, yet only one factor (marital status) appears significant. This should be corrected.

This has been removed from the paragraph

o Lines 271–286 repeat content from lines 251–270. Please revise to avoid redundancy.

Thank you so much for this observation. This was an error and has been rectified

10. Conclusion and Bias Concerns

o The conclusion (line 297) suggests that non-use of emergency contraception is due to access limitations. This is an overgeneralization and could reflect bias. Alternative explanations—such as lack of perceived need, sexual inactivity, or personal preference—should be acknowledged.

o Similarly, line 352 suggests that married women use regular contraception due to planned sexual activity. Yet, if your inclusion criteria excluded those using regular contraceptives, this reasoning may not apply.

Thank you for this observation it is very true . the authors have revised the paragraph as advised

11. Ethical Reporting and Safety Discussion

o The manuscript seems to advocate for proactive and regular use of emergency contraception. If this recommendation is made, information about the safety of frequent use must be included, ideally in the introduction and discussion sections. This has been revised it it has been made clear that “able to access emergency contraceptives whenever necessary to avoid unwanted or unplanned pregnancies”

12. Formatting and Language

o The acknowledgements section (lines 364–365) is incomplete. o Careful proofreading is required to correct grammatical errors and awkward phrasing throughout. This error has been rectified and the section has been completed

---

## [Decision Letter · Decision Letter 2]

19 Feb 2026

Dear Dr. Musinguzi,

Thank you for submitting your manuscript to PLOS ONE. After careful consideration, we feel that it has merit but does not fully meet PLOS ONE’s publication criteria as it currently stands. Therefore, we invite you to submit a revised version of the manuscript that addresses the points raised during the review process.

**1. concerns regarding the quality of written English and the manuscript will benefit from further editing. For instance, line 15 - spelling of "College" written as "Collage"****2. Rewrite the discussion, in particular the first paragraph. Summarize the key findings of your study as it concerns the objectives, and interpret it.****3. Avoid repeating the result as ealier presented unless neccesary.****4. Highlight implication of your findings to policy makers and healthcare professionals****5. Under the limitation, address recall bias.**

We look forward to receiving your revised manuscript.

Kind regards,

Adewale Olufemi Ashimi, MBBS, MPH, PhD, FWACS

Academic Editor

PLOS One

Journal Requirements:

Reviewers' comments:

Reviewer's Responses to Questions

**Comments to the Author**

Reviewer #1: (No Response)

Reviewer #3: (No Response)

2. Is the manuscript technically sound, and do the data support the conclusions?

Reviewer #1: Yes

Reviewer #3: Yes

3. Has the statistical analysis been performed appropriately and rigorously?

Reviewer #1: I Don't Know

Reviewer #3: Yes

4. Have the authors made all data underlying the findings in their manuscript fully available?

Reviewer #1: Yes

Reviewer #3: Yes

5. Is the manuscript presented in an intelligible fashion and written in standard English?

Reviewer #1: No

Reviewer #3: Yes

Reviewer #1: Line 39: The full stop after the SPSS version should be removed

Line 43 and 44: the figures in there should be in parenthesis eg ( 65.98% )

Line 60: Remove the space between year and the full stop and bring the space rather after the full stop

Line 61: unprotected is one word. Kindly correct this and similar ones in the entire manuscript.

Line 90: remove the space before the full stop. Kindly correct this and similar ones in the entire manuscript.

Line 124: Kindly display the formula before assigning values to the variables in the formula.

Line 126: the sentence beginning after the full stop should start with a capital letter (The resulting …..). Kindly correct such and similar ones in the entire manuscript. Line 259 also refers

Line 154: unintended is one word. Kindly correct such and similar words in the entire manuscript.

Line 171: if the variables in the Likert scale started with capital letters, then for consistency sake, strongly disagree should also follow the same pattern. Please correct all such in the entire manuscript. Refer line 206 as well for Residence, table 3 for ‘some emergency…..’

Line 214/215: are the authors suggesting that high doses of POP and COC cannot be used for emergency contraception? Authors should refer to the YUZPE regimen for EC. Kindly address the intent of that statement and let it be clearer.

Table 5: Un married to be unmarried.

Line 259: address the full stop and continue with small letter. Ie ‘Kenya. their’ kindly correct.

Line 326: un intended should be one word

Line 328: change Negative to negative. improve the sentence. The figure cannot be at the end of the sentence in this current construct.

Line 334: should be resource constrained area not areas

Line 296- 306 content has been repeated in Line 307 -318. This was earlier out to authors in Review 1 but the Authors are yet to address it.

Line 323 -325: sentence should be improved

Line 330: change the ‘if’ to ‘of’ this study

I suggest the authors thoroughly proofread the document and improve on the grammar and concordances of the sentences.

Reviewer #3: We'd like to thank the author for revising the manuscript as suggested. All previous comments have been properly assessed, except for minor typo (e.g.: unintended is one word and not: un intended) & grammar errors. Also, in table 2: in the second row, there's an "extra" phrase (emergency contraceptive pill use) that needs removal, I guess.

**Do you want your identity to be public for this peer review?** For information about this choice, including consent withdrawal, please see our For information about this choice, including consent withdrawal, please see our Privacy Policy .

Reviewer #1: No

Reviewer #3: No

---

## [Author Response · Author response to Decision Letter 3]

22 Feb 2026

We sincerely thank the Editor and Reviewer 1 for their careful review of our manuscript and for the constructive comments provided. We greatly appreciate the time and effort invested in evaluating our work. The feedback has been invaluable in improving the clarity, scientific rigor, and overall quality of the manuscript. All comments have been addressed point-by-point below, and corresponding revisions have been incorporated into the revised manuscript. We hope that the changes made have satisfactorily addressed the concerns raised and improved the manuscript to meet the journal’s standards.

Comments Response

Editor

1. concerns regarding the quality of written English and the manuscript will benefit from further editing. For instance, line 15 - spelling of "College" written as "Collage" This has been addressed as reviewers have raised the same concerns, we have done our best to proof read the manuscript and ensure that that the English is improved

Rewrite the discussion, in particular the first paragraph. Summarize the key findings of your study as it concerns the objectives, and interpret it. The discussion section has been rewritten to address all the comments raised

Avoid repeating the result as earlier presented unless necessary. All the repeated results have been removed unless necessary

Highlight implication of your findings to policy makers and healthcare professionals We have directed our implications to the policy makers and health care professionals especially the factors associated

Under the limitation, address recall bias. The limitations paragraph has been rewritten and all the limitations of the study have been elaborated.

Reviewer 1

Line 39: The full stop after the SPSS version should be removed This has been addressed as advised Thank You

Line 43 and 44: the figures in there should be in parenthesis eg (65.98%) Thank you very much for this comment it has been addressed

Line 60: Remove the space between year and the full stop and bring the space rather after the full stop Thank you for this observation. This has been addressed

Line 61: unprotected is one word. Kindly correct this and similar ones in the entire manuscript. This has been addressed thank you

Line 90: remove the space before the full stop. Kindly correct this and similar ones in the entire manuscript.

Thank you for this observation. The comment is addressed

Line 124: Kindly display the formula before assigning values to the variables in the formula. The formula has been added to the manuscript

Line 126: the sentence beginning after the full stop should start with a capital letter (The resulting …..). Kindly correct such and similar ones in the entire manuscript. Line 259 also refers Thank you for this observation this has been corrected

Line 126: the sentence beginning after the full stop should start with a capital letter (The resulting …..). Kindly correct such and similar ones in the entire manuscript. Thank you for this observation this correction has been addressed

Line 259 also refers This has also been addressed as advised thank you

Line 154: unintended is one word. Kindly correct such and similar words in the entire manuscript. Thank you for this observation. This has been corrected

Line 171: if the variables in the Likert scale started with capital letters, then for consistency’s sake, strongly disagree should also follow the same pattern. Please correct all such in the entire manuscript. Refer line 206 as well for Residence, table 3 for ‘some emergency…..’ This has been addressed as advised thank you

Line 214/215: are the authors suggesting that high doses of POP and COC cannot be used for emergency contraception? Authors should refer to the YUZPE regimen for EC. Kindly address the intent of that statement and let it be clearer. This not correctly written and has been revised to ensure that it is correct and the Yuze regiment has been noted

Table 5: Un married to be unmarried. Thank you for this observation. This has been addressed as advised

Line 259: address the full stop and continue with small letter. Ie ‘Kenya. their’ kindly correct. Thank you for this comment this error has been corrected

Line 326: un intended should be one word Thank you for this comment the error has been corrected

Line 328: change Negative to negative. improve the sentence. The figure cannot be at the end of the sentence in this current construct. This has been addressed as advised thank you

Line 334: should be resource constrained area not areas This has been rewritten

Line 296- 306 content has been repeated in Line 307 -318. This was earlier out to authors in Review 1 but the Authors are yet to address it. Thank you very much for understanding and reemphasising this it has been corrected and the repetition removed

Line 323 -325: sentence should be improved Thank you, the whole paragraph has been improved,

Line 330: change the ‘if’ to ‘of’ this study This had been corrected as advised

I suggest the authors thoroughly proofread the document and improve on the grammar and concordances of the sentences. We have done our best to proof read the manuscript Thank you for the advise and the review

---

## [Decision Letter · Decision Letter 3]

12 Mar 2026

Dear Dr. Musinguzi,

Thank you for submitting your manuscript to PLOS ONE. After careful consideration, we feel that it has merit but does not fully meet PLOS ONE’s publication criteria as it currently stands. Therefore, we invite you to submit a revised version of the manuscript that addresses the points raised during the review process.

**The manuscript requires further editing and some sentences need to rewritten. Here are some examples****1.under the abstract, rewrite lines 31-32 or delete some redundant words****2. Line 38; rewrite this or delete quantitative methods in the sentence****3. Line 45; state the mean age with the SD****4. Line 54-55; doesn't make sense. Rewrite or paraphrase it****5. Under background, line 76. Delete "relatively " from the statement****6. Line 84/85, reword the statement and reference low awareness****7.line 90; rewrite this statement either address contraceptive prevalence rates or non use rate to avoid confusion****8. Line 95; "the findings will..." should read "the findings may.."****9. Line 101; delete "quantitative methods " from the statement****10. Reconcile line 90 & line 108. Which one is correct?****11. Under the setting, line 100, provide Information concerning occupation and religion of the residents or inhabitants as they Influence utilization of EC****12.line 139, the  multistage sampling table presented should reflect the estimated sample size of 600 not 585****13. Line 141- Data collection****- who were the research assistants that administered the questionnaire?****- were they trained? By whom? For how long?****- what's their gender?****14. Rewrite line 145****15. Line 194, how was privacy ensured during administration of the questionnaire?****16.Under results, line 205. If 585 respondents were approached and they all participated, the response rate is  100 %. Review this.****17.line 162 - study variables. State clearly who gets 1,2, and 3 points. If the correct answer attracts 3 poi nts, then some respondents may get above 13 with out adequate knowledge using your criterion. This calls for concern, and may send a wrong signal.****18. Under discussion,  line 268. Rewrite the statement " the result of our study ".....****19. Group the utilization of EC in this study and compare with similar studies in the same paragraph. Same applies to knowledge.****20. The lines 293 - 300 should come much earlier in the discussion.****21. No conclusion or recommendation based on the findings****it is important to distinguish between utilization of EC and access to it. Especially in making recommendations.**==============================

We look forward to receiving your revised manuscript.

Kind regards,

Adewale Olufemi Ashimi, MBBS, MPH, PhD, FWACS

Academic Editor

PLOS One

Journal Requirements:

Reviewers' comments:

Reviewer's Responses to Questions

**Comments to the Author**

Reviewer #1: All comments have been addressed

Reviewer #3: All comments have been addressed

2. Is the manuscript technically sound, and do the data support the conclusions?

Reviewer #1: Yes

Reviewer #3: (No Response)

3. Has the statistical analysis been performed appropriately and rigorously?

Reviewer #1: Yes

Reviewer #3: (No Response)

4. Have the authors made all data underlying the findings in their manuscript fully available?

Reviewer #1: Yes

Reviewer #3: (No Response)

5. Is the manuscript presented in an intelligible fashion and written in standard English?

Reviewer #1: Yes

Reviewer #3: (No Response)

Reviewer #1: (No Response)

Reviewer #3: (No Response)

**Do you want your identity to be public for this peer review?** For information about this choice, including consent withdrawal, please see our For information about this choice, including consent withdrawal, please see our Privacy Policy .

Reviewer #1: No

Reviewer #3: No

---

## [Author Response · Author response to Decision Letter 4]

14 Mar 2026

Dear Editor and Reviewers,

We sincerely thank you for your careful review of our manuscript entitled “Knowledge, Attitudes, and Practices Towards Emergency Contraceptive Pill Use Among Women of Reproductive Age in Lira District, Northern Uganda.” We greatly appreciate your constructive comments and valuable suggestions, which have helped us improve the clarity and quality of the manuscript. We have carefully revised the manuscript and addressed all the comments raised. A detailed point-by-point response is provided below, and the corresponding revisions have been made in the manuscript.

We thank you again for your helpful feedback.

Comments Response

1. under the abstract, rewrite lines 31-32 or delete some redundant words Thank you for this comment. We revised the relevant sentences in the abstract to remove redundant wording and improve clarity and conciseness.

2. Line 38; rewrite this or delete quantitative methods in the sentence Thank you for this comment. The sentence was revised and the phrase “quantitative methods” was removed for clarity.

3. Line 45; state the mean age with the SD Thank you for this observation. We revised the results section of the abstract to report the mean age together with the standard deviation as follows: “The mean age of participants was 28.01 ± 7.6 years.”

4. Line 54-55; doesn't make sense. Rewrite or paraphrase it Thank you for this comment. We rephrased the conclusion in the abstract for clarity and readability.

5. Under background, line 76. Delete "relatively" from the statement Thank you. The word “relatively” was deleted as suggested.

6. Line 84/85, reword the statement and reference low awareness Thank you for this important observation. We revised the statement to emphasize the low awareness of emergency contraception among women in Uganda and supported it with the appropriate reference.

7. line 90; rewrite this statement either address contraceptive prevalence rates or non use rate to avoid confusion Thank you. We revised the statement to consistently refer to the contraceptive non-use rate in order to avoid confusion.

8. Line 95; "the findings will..." should read "the findings may.." Thank you. The statement was revised to read “the findings may...” to reflect appropriate caution in interpretation.

9. Line 101; delete "quantitative methods" from the statement Thank you. The phrase “quantitative methods” was deleted and the sentence was revised accordingly.

10. Reconcile line 90 & line 108. Which one is correct? Thank you for this observation. We reconciled the two statements and used the contraceptive non-use rate consistently throughout the manuscript to avoid contradiction.

11. Under the setting, line 100, provide information concerning occupation and religion of the residents or inhabitants as they influence utilization of EC Thank you. We added information on the main occupations and religious composition of residents in Lira District, and briefly linked these contextual factors to possible influence on the utilization of emergency contraception.

12. line 139, the multistage sampling table presented should reflect the estimated sample size of 600 not 585 Thank you for this comment. An additional column was included to show the planned number of respondents to be selected from each village based on the estimated sample size of 600, alongside the final achieved sample of 585.

13. Line 141- Data collection: who were the research assistants that administered the questionnaire? were they trained? By whom? For how long? what's their gender? Thank you for this important comment. We revised the data collection section to clarify that data were collected by five female research assistants who were university graduates. We also added that they received a two-day training conducted by the research team together with a member of the Research Ethics Committee.

14. Rewrite line 145 Thank you. The sentence was rewritten to improve clarity and flow, and now better explains the pretesting of the questionnaire in a comparable setting.

15. Line 194, how was privacy ensured during administration of the questionnaire? Thank you for this comment. We added a statement in the ethical considerations section explaining that interviews were conducted in a private setting within or near the participant’s home, out of hearing range of other household members, and only the participant and research assistant were present.

16. Under results, line 205. If 585 respondents were approached and they all participated, the response rate is 100 %. Review this. Thank you for this observation. We reviewed this section and clarified the wording. The response rate remains 97.5% because the estimated sample size was 600, but 585 respondents were ultimately enrolled and participated.

17. line 162 - study variables. State clearly who gets 1,2, and 3 points. If the correct answer attracts 3 points, then some respondents may get above 13 without adequate knowledge using your criterion. This calls for concern, and may send a wrong signal. Thank you for this valuable comment. We revised the study variables section to clearly state that correct responses were assigned 3 points, partially correct responses 2 points, and incorrect or “don’t know” responses 1 point. We also clarified the score range and explained the rationale for the cutoff used to categorize knowledge levels.

18. Under discussion, line 268. Rewrite the statement "the result of our study"..... Thank you. The wording was revised for better academic tone and clarity.

19. Group the utilization of EC in this study and compare with similar studies in the same paragraph. Same applies to knowledge. Thank you for this suggestion. We reorganized the discussion so that findings on emergency contraceptive utilization are discussed together and compared with similar studies in one paragraph. We did the same for knowledge findings in a separate paragraph.

20. The lines 293 - 300 should come much earlier in the discussion. Thank you. We moved this section earlier in the discussion so that the explanation regarding recent use versus lifetime use appears immediately after the comparison of utilization findings.

21. No conclusion or recommendation based on the findings. It is important to distinguish between utilization of EC and access to it, especially in making recommendations. Thank you for this important observation. We revised the discussion to clearly distinguish between utilization and access to emergency contraception. We also strengthened the concluding part of the discussion by adding clear implications and recommendations based on the findings, with careful wording to avoid conflating use with access.

---

## [Editor Report · Decision Letter 4]

16 Mar 2026

Knowledge, attitudes and practices towards emergency contraceptive pill use among women of reproductive age in Lira District, Northern Uganda: A cross-sectional study

PONE-D-24-31571R4

Dear Mr Musinguzi,

We’re pleased to inform you that your manuscript has been judged scientifically suitable for publication and will be formally accepted for publication once it meets all outstanding technical requirements.

Kind regards,

Adewale Olufemi Ashimi, MBBS, MPH, PhD, FWACS

Academic Editor

PLOS One
---

## [Editor Report · Acceptance letter]

PONE-D-24-31571R4

PLOS One

Dear Dr. Musinguzi,

I'm pleased to inform you that your manuscript has been deemed suitable for publication in PLOS One. Congratulations! Your manuscript is now being handed over to our production team.

Kind regards,

on behalf of

Dr. Adewale Olufemi Ashimi

Academic Editor

PLOS One